# Use of General Health Examination and Cancer Screening among People with Disability Who Need Support from Others: Analysis of the 2016 Comprehensive Survey of Living Conditions in Japan

**DOI:** 10.3390/ijerph21020219

**Published:** 2024-02-13

**Authors:** Takashi Saito, Kumiko Imahashi, Chikako Yamaki

**Affiliations:** 1Department of Social Rehabilitation, Research Institute of National Rehabilitation Center for Persons with Disabilities, 4-1 Namiki, Tokorozawa 359-8555, Japan; imahashi-kumiko@rehab.go.jp; 2Institute for Cancer Control, National Cancer Center, 5-1-1 Tsukiji, Chuou 104-0045, Japan; cyamaki@ncc.go.jp

**Keywords:** general health examination, cancer screening, Japan, disability, disparity

## Abstract

Research on preventive healthcare services among people with disability in Japan is scarce. This study aimed to (1) examine the relationship between disability and the use of general health examination (GHE) and cancer screening (lung, gastric, colorectal, breast and cervical cancer) and (2) explore the reasons for not using GHE. This cross-sectional study used secondary data from individuals aged 20–74 years (*n* = 15,294) from the Comprehensive Survey of Living Conditions of 2016. Binomial logistic regression analysis was conducted to examine the relationship between disability and non-participation in preventive services. In addition, a descriptive analysis was conducted to explore the reasons for non-participation in GHE. Consequently, disability was identified as an independently associated factor for non-participation in GHE (odds ratios (OR): 1.73; 95% confidence interval (95%CI): 1.14–2.62) and screening for colorectal (OR: 1.78; 95%CI: 1.08–2.94), gastric (OR: 2.27; 95%CI: 1.27–4.05), cervical (OR: 2.12; 95%CI: 1.04–4.32) and breast cancer (OR: 2.22; 95%CI: 1.04–4.72), controlling for confounding factors. The most dominant reason for non-participation was “I can go to see the doctor anytime, if I am worried (25/54, 46.3%).” Our findings imply the existence of disability-based disparity in preventive healthcare service use in Japan.

## 1. Introduction

People with disability are more vulnerable to health risks [1,2,3,4], chronic health- [4] and disability-related conditions [5,6] and adverse health-related outcomes [7,8,9] than the general population. This vulnerability, often described as the “narrow margin of health”, [10,11] underlines the importance of the timely use of preventive healthcare services, such as general health examinations and screenings [12,13], for the early detection of a secondary or co-morbid health condition. Some studies [14,15] suggest that the use of preventive healthcare services could influence the achievement of good health-related outcomes for people with disability. Considering the potential benefits, preventive healthcare services can be a promising solution for achieving good health-related outcomes for people with disability as well as the general population.

However, access to preventive healthcare services for people with disability is hindered by various barriers [13]. Increasing evidence has suggested that disability is an important factor that could compromise access to preventive healthcare services [16,17,18,19]. A study from the U.S. [16] reported that major mobility impairments significantly decreased the odds ratios (ORs) for the use of the Papanicolaou test (OR: 0.6; 95% confidence interval (95%CI): 0.4–0.9) and mammography (OR: 0.7; 95%CI: 0.5–0.9) among women, controlled for confounding factors. Similarly, disability was an independent factor associated with the non-use of health examination and flu vaccination in Taiwan [17] and unmet healthcare needs in the UK [18]. According to a report from the World Health Organization (WHO), disparity in access to preventive healthcare services between people with and without disability is a global public health issue [13].

In Japan, general health examinations (GHE) and cancer screening have been common preventive healthcare services since the 1970s. Specifically, under relevant laws, GHE in workplaces (Industrial Health and Safety Act, since 1972) and GHE and cancer screening in municipalities [20] (Health and Medical Service Law for the Aged, since 1983) have been conducted for early detection and treatment and maintaining workers’ and citizens’ health. In 2000, the Japanese government launched a nationwide comprehensive health promotion campaign (National Health Promotion Movement in the 21st Century, called “Health Japan 21”) [21,22], which aimed to prolong healthy life expectancy and improve people’s quality of life. The Japanese government set a goal for the participation rate in GHE and cancer screening and encouraged its citizens to use these preventive healthcare services [21]. Moreover, the government formulated the *Basic Plan to Promote Cancer Control Program* (Fourth edition, in March, 2023) and aimed to establish equal and inclusive cancer screening opportunities for all, including people with disability [23].

However, the accessibility of preventive healthcare services among Japanese people with disability remains understudied. Evidence on this public health issue is still in its infancy in Japan. The number of Japanese people with disability was estimated to be approximately 11 million 600 thousand people as of 2022, which accounted for 9.2% of the entire population [24]. Previous studies [25,26,27,28,29,30,31] have aimed to increase the use of preventive healthcare services. These studies used data from national representative samples and found potential factors associated with the use of preventive healthcare services. However, no factors regarding disability were included in the analysis models and no relevant evidence reported disability-based disparity in access to preventive healthcare services. Some existing evidence [32,33,34,35,36] supported the assumption that use of the preventive services among people with disability might be lower compared to people without disability in Japan. However, these studies focused on a specific preventive service among people with a specific condition, and used hospital-, institution- or peer group-based small convenient samples of people with disability. To date, no studies have focused on disability and examined its relationship with various preventive services among the Japanese population via large-scale national representative data.

Therefore, this study aimed to examine access to preventive healthcare services among people with disability in Japan, and specifically examine the relationship between disability and use of annual GHE and regular cancer screening. In addition, we also aimed to explore the reasons for not using GHE among people with disability. This study sought to bridge the knowledge gap on this issue in Japan and lay the groundwork for further studies that aim to eliminate disparity in preventive health service access among people with disability in Japan. This knowledge could contribute toward enabling policymakers and health practitioners to make informed decisions for addressing this public health issue in Japan. We hypothesized that disability would be an associated factor in not using preventive healthcare services. Owing to this study’s explorational nature, we did not make any hypotheses that corresponded to the second aim.

## 2. Materials and Methods

### 2.1. Study Design

This cross-sectional study conducted a secondary analysis of anonymous and representative data from the Comprehensive Survey of Living Conditions of Japan (CSLC) in 2016. The CSLC dataset was openly provided to researchers and scientists for academic purposes with permission from the Ministry of Health, Labour and Welfare of Japan (MHLW) under Article 36 of the Statistical Act [37]. The MHLW allowed the authors to obtain dataset approval on 11 April 2023.

The CSLC dataset from 2016 was the latest available dataset provided by the MHLW for academic purposes as of when this research was conducted. The CSLC was conducted from 2 June to 14 July 2016. Incidentally, the data did not include those from people who lived in the Kumamoto prefecture, southern Japan (population of 1,785,603 as of 1 January 2016, which accounted for approximately 1.4% of the total Japanese population) owing to a big earthquake in the prefecture in 2016. Hence, the procedure of the survey was compromised. 

Since the authors did not handle any personal data and did not perform any analysis to identify individual persons, this study was exempt from ethical review and informed consent.

### 2.2. Overview of the Preventive Healthcare Service in Japan

The authors focused on two preventive healthcare services: GHE and five types of cancer screening. 

In Japan, annual GHE is provided in workplaces (financed by employers) or in residential places (financed mainly by local municipal government). Typically, annual GHE in the workplace is provided for all the employees and their dependents (such as spouses who are not employed), and in residential places (municipal), is provided for unemployed or self-employed citizens aged 40–74 years and their dependents [38]. 

Cancer screening is also provided in the workplace- or municipal-based screening program. The municipal-based screening program is provided as a nationally established population-based program financed mainly by municipalities. However, employers are not obligated to provide their employees with regular cancer screening. Thus, the availability of workplace-based cancer screening and cost for cancer screening paid by the employees varies based on the workplace. Therefore, generally speaking, Japanese people could use either a workplace- or municipal-based screening program based on their circumstances.

Based on evidence-based guiding principles, the Japanese government recommends five regular cancer screening for individuals who met specific age criteria: gastric cancer screening every two years (those aged 50 years and older), annual lung cancer screening (aged 40 years and older), annual colorectal cancer screening (those aged 40 years and older), breast cancer screening every two years (women aged 40 years and older) and cervical cancer screening every two years (women aged 20 years and older) [38]. No upper age limit has been set by the Japanese government. However, some municipalities have set their own upper age limitation, such as 74 or 79 years, for their municipal-based screening owing to concerns regarding side-effects or accidents relating to the screening [39]. Moreover, the WHO recommends or suggests mammography screening for women aged 40–75 years. However, it provides no recommendations regarding women aged over 75 years [40]. Therefore, in this study, we operationally set 74 years as the upper limit for all five cancer screenings.

Technically, in April 2016, the age criterion for gastric cancer was amended from 40 years and older to 50 years and older. Moreover, in Japan, the recommended frequency for gastric cancer screening varies based on the type of examination individuals receive: an annual X-ray or endoscopic examination every two years.

### 2.3. Data Source

The CSLC has been conducted as a nationwide self-administered cross-sectional survey since 1986 and focuses on the living conditions of Japanese people [41]. The CSLC comprises both large-scale (every three years) and small-scale surveys (annually). The large-scale survey includes questionnaires on household, income, health, saving and long-term care. The small-scale survey only includes questionnaires on household and income. In 2016, a large-scale survey was conducted.

Questionnaires on household and health were distributed to approximately 710,000 households members (290,000 households) from 5410 randomly selected stratified census tracts. Similarly, questionnaires on long-term care and income/saving were distributed to approximately 8000 individuals registered as long-term care users and 80,000 households members (30,000 households) from 2446 and 1963 census tracts which were randomly selected from the aforementioned 5410 census tracts, respectively. All household members, except for those who were temporarily absent during the survey period (e.g., institutionalized individuals), completed the questionnaire. Proxies (such as family members or caregivers) were allowed to answer if study participants could not answer themselves. The response rates for each questionnaire were as follows: household and health (77.5%), long-term care (89.7%) and income and saving (71.8%). Samples of the CSLC questionnaire (in Japanese) are available on the MHLW website [42].

Data provided to researchers and scientists were resampled by the MHLW to maintain anonymity and generalizability. The number of resampling data was comparable to the sampling number of the small-scale survey. To maintain anonymity, some records from households with special circumstances (e.g., households with more than eight members, with more than two members who need any support or supervision from others or with more than two members who are certified as long-term care service user) were removed from the resampling data. Therefore, the presented data or statistics in the current study may not be identical to the official data or statistics from the MHLW. Eventually, the resampled data from 15,294 household members were provided and used in this study. 

### 2.4. Inclusion and Exclusion Criteria

The inclusion criteria included individuals aged 20–74 years. Exclusion criteria included individuals who were temporarily absent from the household during the survey period and those with missing data for any questionnaire items regarding the outcome, explanatory or cofounding variables.

### 2.5. Outcome Variables

The outcome variables were participation or non-participation in GHE and cancer screening. The specific questionnaire items regarding the outcome variables were as follows:Did you receive a GHE in the last year? (Answer option: Yes or No)Did you receive a gastric cancer screening (barium swallow test or endoscopic examination) in the last year? (Answer option: Yes or No). As mentioned earlier, recommended frequency for the gastric cancer varied based on the type of examination: X-ray annually or endoscopic examination every two years. However, in the CSLC in 2016, the questionnaire item only enquired the use of gastric cancer screening in the last year; no questionnaire item enquired of the use of the examination in the last two years. Thus, we used this questionnaire item as an outcome variable for gastric cancer screening.Did you receive a lung cancer screening (chest X-ray or sputum examination) in the last year? (Answer option: Yes or No)Did you receive a colorectal cancer screening (Fecal occult blood tests) in the last year? (Answer option: Yes or No)Did you receive a breast cancer screening (mammography or breast ultrasound) in the last two years? (Answer option: Yes or No)Did you receive a cervical cancer screening (Pap Smear) in the last two years? (Answer option: Yes or No).

### 2.6. Explanatory and Cofounding Variables

The explanatory variable was the status of disability. To operationally define the status of disability, we used a questionnaire item regarding the necessity of support from others in participants’ daily life: Do you need any support or supervision from others due to your disability or declining physical function? (Answer option: Yes or No).We considered individuals to be people with disability if they answered “Yes”.

The use of the preventive healthcare service can be influenced by various factors [13]. To examine the independent association between disability and the use of the services, potential confounding factors should be incorporated into a multivariate analysis model. Thus, we used the items from the CSLC questionnaire which can potentially influence the use of the preventive healthcare service as confounding variables. Candidate confounding variables were selected based on previous studies [16,17,18,25,26,31] that a suggested potential association with participation in GHE or cancer screening. We selected 12 candidate variables from the CSLC questionnaire items. Details of the relevant original questionnaire items (written in Japanese) and their English version (translated by the authors) are available in Appendix A. The candidate variables were divided into three groups: demographic, physiological and psychosocial variables.

Demographic variables included:Sex (male, female).Age. The data for age were categorized into three groups: group one, those who were not eligible for GHE and cancer screening (specifically 20 to 40 or 50 years old, depending on the outcome variables); group 2, those who were eligible for GHE and cancer screening (specifically older than 40 or 50 years old, depended on the outcome variables); group 3, those who were eligible for GHE and cancer screening and 65 years and older. The age of 65 is used as a common threshold of elderly age in policy making as well as the research community in Japan. Thus, we divided the data from individuals eligible for GHE and cancer screening into two groups using the age threshold of 65 years old.Marital status (married, single, divorced/widowed).Educational qualification (primary/junior high school, high school, vocational school/junior college/community (technical) college/university/post-graduate school).


Physiological variables included:
Constant visits to hospitals or clinics, including for dentistry, acupuncture, moxibustion, Japanese massage or Judo therapy (Yes/No). The “constant visit” means regular visit to the hospital or other facilities. Unfortunately, clear definition of frequency (e.g., once a week or once a month) was not described in the questionnaire of the CSLC.Subjective health status (good, normal, bad).Alcohol consumption (never drank or quit drinking, social drinker/low-risk group (>0 to ≤100 g/week); middle-risk drinking (>100 to ≤350 g/week); high-risk drinking (>350 g/week)) [31].Smoking habit (never/ex-smoker, current smoker).

Psychosocial variables included:
Subjective financial state (wealthy; not poor, not wealthy; poor).Kessler Psychological Distress Scale (K6), a measure of mood and anxiety disorder (normal (total score ≤ 4), mild illness (5 to 12), severe illness (13 ≥ total score)) [31].Health insurance (National Health Insurance, employee insurance, other).Employment status (employed, self-employed, employed (other), unemployed).

### 2.7. Reasons of Non-Participation in the General Health Examination

A question regarding non-participation in GHE was followed by a question regarding the reasons: “Why you did not participate your GHE? Select the reasons for non-participation that describe your opinions (multiple choice questions)”. Those who answered “No (non-participation)” to the first question were asked to answer the second question. In total, 12 answer options were provided: (1) I did not know that GHE was available, (2) I was busy, (3) hospitals or venues for the GHE was far, (4) it was costly, (5) I was worried of the examination procedure (blood sampling or endoscopy), (6) I was admitted in hospital when the GHE was available, (7) I did not think annual GHE was necessary, (8) I did not need the GHE because I am healthy, (9) I can go to see the doctor anytime, if I am worried, (10) I was worried of results of the GHE, (11) I was too lazy to participate or (12) other. Identical answer options have been applied in the CSLC since 1998. We used this questionnaire item to explore the reasons for non-participation. Unfortunately, in the CSLC questionnaire, there was no following question regarding reasons for non-participation in cancer screening.

### 2.8. Statistical Analysis

All variables were categorical and are expressed in numbers and percentages. 

First, a chi-squared test was conducted to examine the differences in explanatory and confounding variables between participants and non-participants in GHE and cancer screening. Second, unadjusted ORs for non-participation were calculated for each explanatory and confounding variable. Third, a binomial logistic regression analysis that used the forced entry method was performed to identify the factors associated with non-participation. We developed an analysis model with status of disability as an explanatory variable and all candidate confounding variables as the covariates. No confounding variables that were highly correlated with each other were observed in the analysis model. Specifically, the phi coefficient (for a confounding variable with a two-by-two contingency table) and Cramer’s coefficient of association (for a confounding variable with a three or more-by-three or more contingency table) between the confounding variables were less than 0.5.

Finally, a descriptive analysis was conducted to explore the reasons for non-participation.

All analyses were performed using IBM SPSS (version 28.0.1.0). Statistical significance level was set at *p* < 0.05.

## 3. Results

Figure 1 describes the participant selection process. After data that met the exclusion criteria were removed, we identified the eligible data for GHE (those aged 20–74 years, *n* = 8438) and gastric (those aged 50–74 years, *n* = 4318), lung (those aged 40–74 years, *n* = 6042), colorectal (those aged 40–74 years, *n* = 6030), breast (women aged 40–74 years, *n* = 3098) and cervical cancer screenings (women aged 20–74 years, *n* = 4261). Details of the number of individuals with missing data are shown in Appendix A.

Table 1 present the participants’ characteristics and their comparison between participants and non-participants in preventive healthcare services, respectively. Overall, the participation rates in GHE and lung, colorectal, gastric, cervical and breast cancer screenings were 73.4% (6192/8438), 52.4% (3163/6042), 46.8% (2825/6030), 47.1% (2033/4318), 43.7% (1864/4261) and 47.4% (1469/3098), respectively. Similarly, among individuals with disability, the participation rates were 50.0% (54/108), 32.6% (28/86), 28.2% (24/85), 25.0% (17/68), 20.8% (11/53) and 25.0% (10/40), respectively. Individuals with disability showed significantly higher non-participation rates for general health examination and all cancer screening (*p* < 0.001 or *p* = 0.004) than their counterparts. Of the 12 confounding factors, eight showed significant differences between participants and non-participants in GHE and all cancer screening.

Table 2 presents the results of the binomial logistic regression analysis. The adjusted ORs were calculated after controlling for all confounding variables. Significant positive associations were observed between disability and non-participation in GHE (OR: 1.73; 95%CI: 1.14–2.62) and colorectal (OR: 1.78; 95%CI: 1.08–2.94), gastric (OR: 2.27; 95%CI: 1.27–4.05), cervical (OR: 2.12; 95%CI: 1.04–4.32) and breast cancer screenings (OR: 2.22; 95%CI: 1.04–4.72), controlling for the confounding factors. No statistically significant association was observed between disability and non-participation in lung cancer screening (OR: 1.56; 95%CI: 0.96–2.51), controlling for confounding factors. The results of the analysis for the other confounding variables can be found in the Appendix A.

Figure 2 presents the reasons for non-participation in GHE. The three most dominant reasons (excluding “other”) were: I can go to see the doctor anytime, if I am worried (25/54, 46.3%), I was admitted in hospital when the GHE was available (12/54, 22.2%) and I was too lazy to participate (8/54, 14.8%).

## 4. Discussion

### 4.1. Key Points 

We analyzed data from the CSLC in 2016, a nationwide representative sample of the Japanese population, and revealed an association between disability and non-participation in GHE and four types of cancer screening. These findings implied that disability-based disparity may exist in preventive healthcare service use in Japan. To the best of our knowledge, this was the first study that highlighted this public health issue via a large-scale representative sample. We also found some barriers that Japanese people with disability faced when using GHE. These findings could bridge the knowledge gap in this public health issue and shed light on the necessity to tackle it in Japan. Our study provides suggestions that can lay the groundwork for future studies aiming to eliminate disability-based disparity in preventive healthcare use in Japan.

### 4.2. Association between Disability and Preventive Healthcare Service Use in Japan

Our study showed that disability was a significantly associated factor for non-participation in GHE and the screening of gastric, colorectal, breast and cervical cancer, controlling for 12 confounding factors. This finding implied that disability-based disparity may exist in preventive healthcare service use in Japan. This was our most notable finding.

Our findings were consistent with those of previous articles that examined disability as an associated factor for preventive healthcare service use. Andiwijaya et al. [19] conducted a systematic review and meta-analysis and concluded that women with disability faced disparities in the reception of breast cancer screening (pooled adjusted OR: 0.78; (95%CI: 0.72, 0.85) and cervical cancer screening (pooled adjusted OR: 0.67; 95%CI: 0.47, 0.94) compared to women without disability. Moreover, a study [43] analyzed cancer registration data (93,545 records from patients diagnosed with any type of cancer) from principal hospitals in northwestern Japan and reported that there was a significant difference in pathways for the detection of stage 0 or stage 1 cancer between people with and without disability. Specifically, people with disability with stage 0 or 1 cancer were diagnosed mainly through their regular hospital visits (44.9% and 54.5%, respectively), not through screening (14.7% and 9.4%, respectively). Conversely, their counterparts (people without disability) were diagnosed through cancer screening (31.4% and 24.2%) or regular hospital visits (34.3% and 39.6%, respectively). The study showed indirect evidence that the use of cancer screening could be disproportionately compromised owing to disability in Japan. Our findings supports the assumptions of previous studies and highlights the disparities in the use of preventive healthcare services in Japan.

We found no statistically significant association between disability and non-participation in lung cancer screening, which was unexpected. Admittedly, findings on disability as a potential barrier to for lung cancer screening use were mixed. A scoping review [44] that focuses on barriers and facilitators to lung cancer screening use in the US reported supportive evidence that a patient’s “comorbidity” could be a potential barrier. However, another scoping review [45] focused on behavioral barriers and facilitators to lung cancer screening from high-income countries and reported no disability-related behavioral barriers. Although the exact reasons were unclear, we speculated two possible explanations as to why only lung cancer screening was not associated with disability in our study. First, the procedure of lung cancer screening in Japan, specifically chest X-ray or sputum examination, might be easier or more acceptable for people with disability than other types of cancer screening. This might enable lung cancer screening to be more accessible for people with disability. Anecdotal and qualitative evidence [46,47] suggested that an individual with physical disability would face difficulties in undergoing cervical (Papanicolaou test) and breast cancer tests (mammography tests) as they require certain movements, such as moving on or transferring to a high examination table. Gastric (barium swallow test or endoscopic examination) and colorectal cancer screenings (fecal occult blood tests) may also involve some difficulty with procedures for people with disability during their examination. Conversely, lung cancer screening may involve relatively simple procedures, such as just standing in front of an X-ray machine or providing a sputum specimen. These relatively simple procedures might be attributable to the lack of an association between disability and non-participation in our study. Second, faulty recollection and overestimation of the experience of undergoing lung cancer screening might be factors in this lack of association. Lung cancer screening in Japan generally includes chest X-ray or sputum examination. These examinations, especially chest X-ray, are common examinations and are conducted for various purposes other than cancer screening (such as examination for pneumonia). Moreover, it is mandatory for some people who work in schools, hospitals or nursing care facilities to take an annual examination for the early detection of tuberculosis and prevention of its mass infection. Examinations for tuberculosis also include a chest X-ray. Therefore, individuals who worked in these facilities and underwent chest X-rays not aimed at cancer screening may have misunderstood and responded that they underwent lung cancer screening in the last year. Given that the CSLC was a self-reported questionnaire survey, misclassification could have occurred and influenced the results.

### 4.3. Reasons for Non-Participation in the General Health Examination

It is crucial to determine the reasons for non-participation to specify the barriers people with disability face and to envisage effective strategies. Therefore, we explored these reasons via the questionnaire item in the CSLC.

The two most dominant reasons were: I can go to see the doctor anytime, if I am worried (25/54, 46.3%) and I was admitted in hospital when the GHE was available (12/54, 22.2%). Thus, we conjectured that there might be some misconceptions regarding GHE among people with disability. First, even without any noticeable symptoms, GHE should be taken regularly for the early detection of diseases. If an individual realized their symptoms, felt anxious and went to see the doctor, it might be too late. Second, generally speaking, GHE was available anytime as a workplace-based or municipality-based examination in Japan. Even if there were periods of hospital admission, this would not compromise their opportunities to use GHE. If these misconceptions played roles as barriers that hindered access to GHE, campaigns or activities that aimed to spread accurate information and raise awareness regarding the health examination could be a solution to increase the participation rate among people with disability in Japan. 

The third common reason (other than “other”) was I was too lazy to participate. Why were people with disability reluctant to participate in GHE? Potential barriers could include difficulty searching for an examination venue to accommodate people with disability, long waiting times, difficulty undergoing examination procedures and negative past experiences owing to disability-related issues. However, the specific barriers were unclear owing to the ambiguity of this option. Unfortunately, this ambiguity did not allow us to specify the barriers and envisage specific solutions. The necessity to enable specific answer options in the CSLC was implied.

The three most common reasons in this study included no reasons relating to the geographic/physical accessibility or affordability issues. These reasons were placed at a relatively at lower rank, fifth for affordability-related issues (“It was costly”, 5.6%) and ninth for geographic/physical accessibility-related issues (“Hospital for the examination was far”, 3.7%), respectively. The challenges or difficulties relating to geographic/physical accessibility or affordability, however, are reported to be common barriers for people with disability to access healthcare services [13], especially in less resourced areas [48,49,50,51]. Maart S. [48], for instance, investigated the use of health services among 152 people with disability living in a low-income area in South Africa. Their findings were that the main problems with accessing services included inadequate finances (71%) and transport problems (72%). Moreover, a literature review [49] of 50 eligibles articles which examined access to general healthcare services for people with disability in low- and middle-income countries reported that “transport difficulties” and “financial difficulties”, as well as the “attitudes of staff”, were the most commonly reported barriers across the review articles. Contrary, a report from a well-resourced country, Australia [50], indicated that difficulties or problems relating to cost or physical accessibility (3.6 to 12%) were not necessarily the most common issues with using healthcare services among people with disability. Instead, the issue of “unacceptable or lengthy waiting times” (24 to 31%) was a more common difficulty for them. Although a direct comparison between the findings of current study and the existing studies is impossible due to the heterogeneity of the study design and targeted health services, our findings were seemingly similar to the findings from Australia [50], a well-resourced country. Although the geographic/physical accessibility or affordability issues should not be ignored, their relative importance or priority may be lower than other issues in a well-resourced country like Japan.

Last but not least, the answer option “other” was also a common answer option chosen in this study. Unsurprisingly, no concreate solutions could be made from this answer. This implied that the answer options in the CSLC questionnaire might not fully cover the barriers people with disability face and necessities the creation of more comprehensive answer options. 

### 4.4. Implication for Future Studies Aiming to Eliminate Disability-Based Disparity in Preventive Service Utilization

This study implies that disability-based disparity may exist in preventive healthcare service use in Japan. This finding sheds light on the necessity to tackle this public health issue. We believe that our implications can contribute toward laying the groundwork for future studies aiming to eliminate disability-based disparity in preventive healthcare use in Japan.

First, our definition of disability was related to a self-reported necessity for support or supervision from others. This self-report-based definition may lead to misclassification, impacting on our study findings. Specifically, this definition may reflect some typical conditions, such as severe physical or intellectual disability. However, those who have other types of disabilities who do not necessarily need support from others, such as hearing impairment, vision impairment, minor developmental or physical disability, might not be represented in our definition. Moreover, those who need support but practically refuse or are unable to acknowledge it were not included as people with disability in this study, leading to underestimation. Contrarily, those who are healthy but temporally need some support due to temporary disabling conditions (e.g., fracture of the leg) might be included as people with disability, leading to over-estimation. This issue of misclassification was also reported in a previous study [16] which used self-administered disability data. Although, our study findings implies a necessity to expand the research scope to a wider range of disabilities in further research, the disability-related data ideally should be corrected through reliable and validated data sources, such as a registry data source that includes diagnostic data on disability provided by medical professionals. 

Second, although data from the CSLC provided good opportunities for researchers to examine the relationships between relevant factors, our study identified some drawbacks of the CSLC dataset for exploring reasons for non-participation. These drawbacks prevented us from obtaining relevant information on the barriers to GHE use among people with disability. Understanding specific and comprehensive reasons for non-participation in preventive healthcare services is crucial for creating concrete and effective solutions. Further studies aiming to specifically and comprehensively explore the reasons for non-participation in preventive healthcare services are necessary. Existing representative models [52,53,54] of the associated factors for health service access may be helpful to gain a better understanding of specific and comprehensive barriers to preventive healthcare service use among people with disability in Japan.

### 4.5. Study Limitations

This study has some limitations. First, as mentioned earlier, the misclassification of data regarding the use of preventive healthcare services and the state of disability may have influenced the study results owing to the self-administered nature of the CSLC. Second, we excluded participants aged 75 years or older and those who had missing data in our analysis, which may have affected our results. Third, proxies’ answer may have influenced the results. In the CSLC, proxies were allowed to complete the questionnaire when the study participant could not answer themselves. While the accuracy of the percentages of responses written by proxies was unclear, respondents with disability were likely to have asked for support from their proxies. The proxies’ involvement may have influenced the results. Fourth, a huge gap betwsuppleen the number of participants with and without disability was observed in this study. This gap may have influenced the results of our statistical analysis. Fifth, the data from the CSLC in 2016 were the latest data we could obtain. However, this was not the latest dataset. Moreover, in 2016, there was no COVID-19 pandemic, which could have profoundly influenced access to healthcare services [55]. Hence, for assuring the adaptability of their findings to the current situation in Japan, further studies should use the most recent data.

## 5. Conclusions

This study used a nationwide representative dataset of Japan and demonstrated an association between disability and non-participation in GHE and four types of cancer screenings. This finding implied that disability-based disparity may exist in preventive healthcare service use in Japan. These findings can bridge the knowledge gap on the public health issue and shed light on the necessity to tackle this health issue in Japan. Our implications could contribute to laying the groundwork for future studies aiming to eliminate disability-based disparity in preventive healthcare utilization in Japan.

## Figures and Tables

**Figure 1 ijerph-21-00219-f001:**
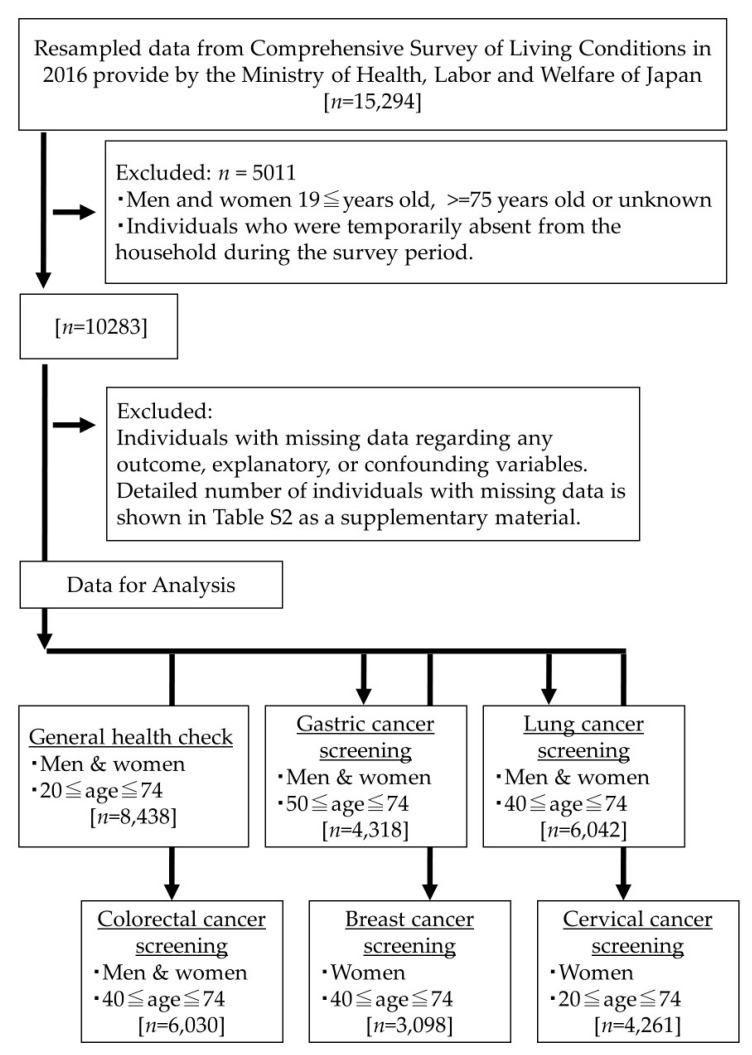
Participant selection process.

**Figure 2 ijerph-21-00219-f002:**
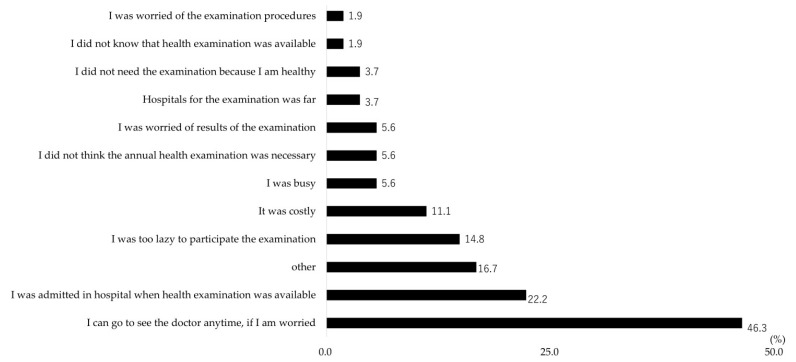
Reasons for non-participation in general health examination among people with disability who did not participate general health examination (*n* = 54). The question on the reasons for non-participation was provided as a multiple choice question.

**Table 1 ijerph-21-00219-t001:** Participants’ characteristics and their comparison between participants and non-participants in preventive healthcare services.

	General Health Examination (*n* = 8438)					Lung Cancer Screening (*n* = 6042)					Colorectal Cancer Screening (*n* = 6030)				
	Participants (*n* = 6196)		Non- Participants (*n* = 2242)		*p*-Value	Participants (*n* = 3163)		Non- Participants (*n* = 2879)		*p*-Value	Participants (*n* = 2825)		**Non-** **Participants** **(*n* = 3205)**		** *p* ** **-Value**
	*n*	(%)	*n*	(%)		*n*	(%)	*n*	(%)		*n*	(%)	** *n* **	**(%)**	
Disability					<0.001					<0.001					<0.001
Do not need any support or supervision	6142	(99.13)	2188	(97.59)		3135	(99.11)	2821	(97.99)		2801	(99.15)	3144	(98.10)	
Need support or supervision	54	(0.87)	54	(2.41)		28	(0.89)	58	(2.01)		24	(0.85)	61	(1.90)	
Sex					<0.001					<0.001					<0.001
Male	3229	(52.11)	880	(39.25)		1720	(54.38)	1231	(42.76)		1510	(53.45)	1421	(44.34)	
Female	2967	(47.89)	1362	(60.75)		1443	(45.62)	1648	(57.24)		1315	(46.55)	1784	(55.66)	
Age (years)					<0.001					0.032					0.332
65–74	1264	(20.40)	576	(25.69)		914	(28.90)	905	(31.43)		834	(29.52)	983	(30.67)	
40–64	3325	(53.66)	982	(43.80)		2249	(71.10)	1974	(68.57)		1991	(70.48)	2222	(69.33)	
20–39	1607	(25.94)	684	(30.51)		-	(-)	-	(-)		-	(-)	-	(-)	
Marital status					<0.001					<0.001					<0.001
Married	4395	(70.93)	1439	(64.18)		2594	(82.01)	2189	(76.03)		2355	(83.36)	2425	(75.66)	
Single	1306	(21.08)	585	(26.09)		273	(8.63)	346	(12.02)		221	(7.82)	391	(12.20)	
Divorced/widowed	495	(7.99)	218	(9.72)		296	(9.36)	344	(11.95)		249	(8.81)	389	(12.14)	
Educational qualification					<0.001					<0.001					<0.001
Vocational school/junior college/community (technical) college/university/post-graduate school	3278	(52.91)	911	(40.63)		1533	(48.47)	1108	(38.49)		1419	(50.23)	1224	(38.19)	
High school	2512	(40.54)	1025	(45.72)		1369	(43.28)	1413	(49.08)		1199	(42.44)	1576	(49.17)	
Primary/junior high school	406	(6.55)	306	(13.65)		261	(8.25)	358	(12.43)		207	(7.33)	405	(12.64)	
Subjective financial state					<0.001					<0.001					<0.001
Wealthy	391	(6.31)	85	(3.79)		241	(7.62)	117	(4.06)		227	(8.04)	132	(4.12)	
Nor poor, not wealthy	2492	(40.22)	723	(32.25)		1323	(41.83)	1018	(35.36)		1203	(42.58)	1138	(35.51)	
Poor	3313	(53.47)	1434	(63.96)		1599	(50.55)	1744	(60.58)		1395	(49.38)	1935	(60.37)	
Health insurance					<0.001					<0.001					<0.001
Employee insurance	4519	(72.93)	1151	(51.34)		2132	(67.40)	1517	(52.69)		1866	(66.05)	1780	(55.54)	
National Health Insurance	1590	(25.66)	995	(44.38)		985	(31.14)	1249	(43.38)		917	(32.46)	1310	(40.87)	
Other	87	(1.40)	96	(4.28)		46	(1.45)	113	(3.92)		42	(1.49)	115	(3.59)	
Employment status					<0.001					<0.001					<0.001
Employed	4058	(65.49)	845	(37.69)		1867	(59.03)	1228	(42.65)		1584	(56.07)	1504	(46.93)	
Self-employed	333	(5.37)	218	(9.72)		219	(6.92)	268	(9.31)		194	(6.87)	291	(9.08)	
Employed (other)	400	(6.46)	193	(8.61)		267	(8.44)	246	(8.54)		244	(8.64)	270	(8.42)	
Unemployed	1405	(22.68)	986	(43.98)		810	(25.61)	1137	(39.49)		803	(28.42)	1140	(35.57)	
Kessler Psychological Distress Scale					0.002					0.007					<0.001
Normal (total score ≤ 4)	4510	(72.79)	1549	(69.09)		2376	(75.12)	2068	(71.83)		2139	(75.72)	2295	(71.61)	
Mild illness (5 ≤ total score ≤ 12)	1468	(23.69)	591	(26.36)		705	(22.29)	711	(24.70)		614	(21.73)	801	(24.99)	
Severe illness (13 ≤ total score)	218	(3.52)	102	(4.55)		82	(2.59)	100	(3.47)		72	(2.55)	109	(3.40)	
Constant visits to hospitals ^†^					<0.001					<0.001					<0.001
Yes (visit)	2668	(43.06)	858	(38.27)		1653	(52.26)	1341	(46.58)		1527	(54.05)	1467	(45.77)	
No (no visit)	3528	(56.94)	1384	(61.73)		1510	(47.74)	1538	(53.42)		1298	(45.95)	1738	(54.23)	
Subjective health status					<0.001					0.002					0.024
Good	2288	(36.93)	781	(34.83)		1089	(34.43)	931	(32.34)		971	(34.37)	1043	(32.54)	
Normal	3262	(52.65)	1158	(51.65)		1731	(54.73)	1552	(53.91)		1540	(54.51)	1735	(54.13)	
Bad	646	(10.43)	303	(13.51)		343	(10.84)	396	(13.75)		314	(11.12)	427	(13.32)	
Alcohol consumption					<0.001					<0.001					<0.001
Never or quit drinking	3197	(51.60)	1410	(62.89)		1529	(48.34)	1704	(59.19)		1373	(48.60)	1856	(57.91)	
Social drinker/low-risk group (>0 to ≤100 g/week)	1534	(24.76)	444	(19.80)		776	(24.53)	564	(19.59)		708	(25.06)	627	(19.56)	
Middle-risk drinking (>100 to ≤350 g/week)	1278	(20.63)	322	(14.36)		765	(24.19)	520	(18.06)		655	(23.19)	629	(19.63)	
High-risk drinking (>350 g/week)	187	(3.02)	66	(2.94)		93	(2.94)	91	(3.16)		89	(3.15)	93	(2.90)	
Smoking habit					0.157					0.058					<0.001
Never/ex-smoker	4881	(78.78)	1734	(77.34)		2507	(79.26)	2224	(77.25)		2301	(81.45)	2429	(75.79)	
Current smoker	1315	(21.22)	508	(22.66)		656	(20.74)	655	(22.75)		524	(18.55)	776	(24.21)	
	**Gastric Cancer Screening** **(*n* = 4318)**					**Cervical Cancer Screening** **(*n* = 4261)**					**Breast Cancer Screening** **(*n* = 3098)**				
	**Participants** **(*n* = 2033)**		**Non-** **Participants** **(*n* = 2285)**		***p*-Value**	**Participants** **(*n* = 1864)**		**Non-** **Participants** **(*n* = 2397)**		** *p* ** **-Value**	**Participants** **(*n* = 1469)**		**Non-** **Participants** **(*n* = 1629)**		** *p* ** **-Value**
	** *n* **	**(%)**	** *n* **	**(%)**		** *n* **	**(%)**	** *n* **	**(%)**		** *n* **	**(%)**	** *n* **	**(%)**	
Disability					<0.001					<0.001					0.004
Do not need any support or supervision	2016	(99.16)	2234	(97.77)		1853	(99.41)	2355	(98.25)		1459	(99.32)	1599	(98.16)	
Need support or supervision	17	(0.84)	51	(2.23)		11	(0.59)	42	(1.75)		10	(0.68)	30	(1.84)	
Sex					<0.001					-					-
Male	1106	(54.40)	1003	(43.89)		-	(-)	-	(-)		-	(-)	-	(-)	
Female	927	(45.60)	1282	(56.11)		1864	(100.00)	2397	(100.00)		1469	(100.00)	1629	(100.00)	
Age (years)					0.033					<0.001					<0.001
65–74	-	(-)	-	(-)		284	(15.24)	629	(26.24)		325	(22.12)	590	(36.22)	
40–64	-	(-)	-	(-)		1096	(58.80)	1075	(44.85)		1144	(77.88)	1039	(63.78)	
20–39	-	(-)	-	(-)		484	(25.97)	693	(28.91)		-	(-)	-	(-)	
65–74	821	(40.38)	996	(43.59)		-	(-)	-	(-)		-	(-)	-	(-)	
50–64	1212	(59.62)	1289	(56.41)		-	(-)	-	(-)		-	(-)	-	(-)	
Marital status					<0.001					<0.001					<0.001
Married	1705	(83.87)	1757	(76.89)		1451	(77.84)	1491	(62.20)		1195	(81.35)	1222	(75.02)	
Single	124	(6.10)	207	(9.06)		221	(11.86)	588	(24.53)		94	(6.40)	125	(7.67)	
Divorced/widowed	204	(10.03)	321	(14.05)		192	(10.30)	318	(13.27)		180	(12.25)	282	(17.31)	
Educational qualification					<0.001					<0.001					<0.001
Vocational school/junior college/community (technical) college/university/post-graduate school	912	(44.86)	767	(33.57)		1015	(54.45)	1080	(45.06)		722	(49.15)	573	(35.17)	
High school	921	(45.30)	1173	(51.33)		765	(41.04)	1061	(44.26)		660	(44.93)	842	(51.69)	
Primary/junior high school	200	(9.84)	345	(15.10)		84	(4.51)	256	(10.68)		87	(5.92)	214	(13.14)	
Subjective financial state					<0.001					<0.001					<0.001
Wealthy	175	(8.61)	84	(3.68)		137	(7.35)	93	(3.88)		118	(8.03)	58	(3.56)	
Not poor, not wealthy	868	(42.70)	853	(37.33)		772	(41.42)	838	(34.96)		611	(41.59)	592	(36.34)	
Poor	990	(48.70)	1348	(58.99)		955	(51.23)	1466	(61.16)		740	(50.37)	979	(60.10)	
Health insurance					<0.001					<0.001					<0.001
Employee insurance	1188	(58.44)	1001	(43.81)		1399	(75.05)	1465	(61.12)		1003	(68.28)	855	(52.49)	
National Health Insurance	810	(39.84)	1185	(51.86)		452	(24.25)	862	(35.96)		455	(30.97)	717	(44.01)	
Other	35	(1.72)	99	(4.33)		13	(0.70)	70	(2.92)		11	(0.75)	57	(3.50)	
Employment status					<0.001					<0.001					<0.001
Employed	992	(48.79)	822	(35.97)		1070	(57.40)	1195	(49.85)		777	(52.89)	652	(40.02)	
Self-employed	168	(8.26)	231	(10.11)		44	(2.36)	64	(2.67)		40	(2.72)	54	(3.31)	
Employed (other)	184	(9.05)	218	(9.54)		130	(6.97)	179	(7.47)		122	(8.30)	159	(9.76)	
Unemployed	689	(33.89)	1014	(44.38)		620	(33.26)	959	(40.01)		530	(36.08)	764	(46.90)	
Kessler Psychological Distress Scale					0.063					0.931					0.627
Normal (total score ≤ 4)	1558	(76.64)	1681	(73.57)		1302	(69.85)	1685	(70.30)		1040	(70.80)	1178	(72.31)	
Mild illness (5 ≤ total score ≤ 12)	423	(20.81)	542	(23.72)		483	(25.91)	609	(25.41)		378	(25.73)	395	(24.25)	
Severe illness (13 ≤ total score)	52	(2.56)	62	(2.71)		79	(4.24)	103	(4.30)		51	(3.47)	56	(3.44)	
Constant visits to hospitals ^†^					<0.001					0.004					0.001
Yes (visit)	1249	(61.44)	1225	(53.61)		848	(45.49)	985	(41.09)		782	(53.23)	771	(47.33)	
No (no visit)	784	(38.56)	1060	(46.39)		1016	(54.51)	1412	(58.91)		687	(46.77)	858	(52.67)	
Subjective health status					0.099					0.023					0.495
Good	651	(32.02)	677	(29.63)		713	(38.25)	820	(34.21)		508	(34.58)	531	(32.60)	
Normal	1134	(55.78)	1289	(56.41)		940	(50.43)	1296	(54.07)		784	(53.37)	892	(54.76)	
Bad	248	(12.20)	319	(13.96)		211	(11.32)	281	(11.72)		177	(12.05)	206	(12.65)	
Alcohol consumption					<0.001					0.008					0.004
Never or quit drinking	1001	(49.24)	1340	(58.64)		1230	(65.99)	1690	(70.50)		984	(66.98)	1178	(72.31)	
Social drinker/low-risk group (>0 to ≤100 g/week)	504	(24.79)	437	(19.12)		433	(23.23)	507	(21.15)		345	(23.49)	298	(18.29)	
Middle-risk drinking (>100 to ≤350 g/week)	478	(23.51)	448	(19.61)		178	(9.55)	176	(7.34)		125	(8.51)	137	(8.41)	
High-risk drinking (>350 g/week)	50	(2.46)	60	(2.63)		23	(1.23)	24	(1.00)		15	(1.02)	16	(0.98)	
Smoking habit					<0.001					0.003					<0.001
Never/ex-smoker	1692	(83.23)	1790	(78.34)		1708	(91.63)	2130	(88.86)		1368	(93.12)	1435	(88.09)	
Current smoker	341	(16.77)	495	(21.66)		156	(8.37)	267	(11.14)		101	(6.88)	194	(11.91)	

^†^ No clear definition regarding the frequency of visit is provided in the questionnaire.

**Table 2 ijerph-21-00219-t002:** Unadjusted and adjusted odds ratios of disability for non-participation in general health examination and cancer screening.

	General Health Examination		Cancer Screening									
			Lung		Colorectal		Gastric		Cervical		Breast	
	Unadjusted OR (95%CI)	Adjusted OR (95%CI)	Unadjusted OR (95%CI)	Adjusted OR (95%CI)	Unadjusted OR (95%CI)	Adjusted OR (95%CI)	Unadjusted OR (95%CI)	Adjusted OR (95%CI)	Unadjusted OR (95%CI)	Adjusted OR (95%CI)	Unadjusted OR (95%CI)	Adjusted OR (95%CI)
Disability												
Do not need any support or supervision	1 (Ref)	1 (Ref)	1 (Ref)	1 (Ref)	1 (Ref)	1 (Ref)	1 (Ref)	1 (Ref)	1 (Ref)	1 (Ref)	1 (Ref)	1 (Ref)
Need support or supervision	2.81 (1.92, 4.11)	1.73 (1.14, 2.62)	2.30 (1.46, 3.62)	1.56 (0.96, 2.51)	2.26 (1.40, 3.64)	1.78 (1.08, 2.94)	2.71 (1.56, 4.70)	2.27 (1.27, 4.05)	3.00 (1.54, 5.85)	2.12 (1.04, 4.32)	2.74 (1.33, 5.62)	2.22 (1.04, 4.72)

Note: OR, odds ratio; CI, confidence interval; Ref, reference. For adjusted OR, confounding variables controlled were: sex, age, marital status, education qualification, constant visits to hospital, subjective health status, alcohol consumption, smoking habit, subjective financial state, Kessler Psychological Distress Scale, health insurance and employment status.

## Data Availability

No new data were created or analyzed in this study. Data sharing is not applicable to this study.

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
