# Peer review of "Use of General Health Examination and Cancer Screening among People with Disability Who Need Support from Others: Analysis of the 2016 Comprehensive Survey of Living Conditions in Japan"

_ijerph, 2024, doi:10.3390/ijerph21020219_

Round 1

Reviewer 1 Report

Comments and Suggestions for Authors

The proposed study “Disabilities and use of general health examination and cancer screening in Japan: analysis of the 2016 comprehensive survey of living conditions” presents a statistical analysis on a widely administered (n = 15’294) survey for Japan population, regarding general living conditions. From this source, authors apply a binomial logistic regression analysis with forced entry method to assess if the condition of disability (measured by the self-reported need for daily support from other people) has an impact on general health screening and some cancer screenings, verifying that people with disability has a lower level of adoption of such preventive measures.

The study's methodology appears to rely on conventional approaches, and there may be an opportunity to enhance the originality of the research by exploring more innovative methods. However, the exposition is very clear and scientifically sound.

Regardless the methodology, I only have a limited number of comments:

1) Methods section: I suggest specifying here what strategy (and related metrics) is applied in order to assess the impact of the target variable (disability) on the model over other confounding factors, or in this case why the chosen model and parameters allows to do that.

2) Check resolution of figure 1.

3) Results section: I suggest adding a brief presentation of most noticeable results (from a quantitative point of view), thus helping the reader to start focusing on more relevant aspects.

4) Discussion section: I suggest adding more complete numerical references to the relevant results discussed.

Comments on the Quality of English Language

Minor typos to be checked.

Author Response

Reply to reviewer 1

Thank you very much for your review and for your helpful comments. We have revised this manuscript on the basis of your comments for us. Revisions are highlighted in yellow color in the manuscript. Please find attached file. 

Reviewer 2 Report

Comments and Suggestions for Authors

Stated purpose of this manuscript was to examine relationships between disabilities and preventive health screenings, especially as related to cancer screenings.  This paper is generally well-written and has value regarding contributions to available literature.  There are several modifications needed to this manuscript prior to being ready for publication, however:

1.       Although the writing in the manuscript is generally good, there are several places where there are typographical/grammatical/spelling errors (e.g., line 86 – “furthers studies”; line 212 – “qualificaiotn”.  The authors need to review the manuscript and correct those errors.

2.       An overall weakness of the paper is the age of the data (i.e., 2016), but the authors explain the reasoning behind use of that particular dataset; thus, this is not the issue.  The concern is that collection of this data commenced and ended prior to the onset of the COVID pandemic.  There needs to be some mention in the discussion about this, especially because there is clear data revealing impact of the COVID pandemic on health screenings overall.  And there is strong recognition of the differential impact of the COVID pandemic on individuals with a disability.  The authors need to highlight this in the discussion.  In many ways, that could strengthen their argument that disability status impacts health screenings (but also see point 6).

3.       On lines 107-108, the authors mentioned that ethical review and informed consent was not necessary because there was no handling of personal data.  Although informed consent is likely not necessary, most institutions still require some sort of institutional review of any study utilizing data from human participants (though I am not familiar with specific requirements in Japan).  If such a process review is necessary, this type of study is likely what might qualify for the label of Exempt (or something analogous). 

4.       The authors do a nice job with the introduction and provide a strong justification for the general study. What is missing, however, is justification for limiting their analyses to cancer.  Although limiting their study to specific questions is good thing (doing more than they did would have been too much), the authors do need some justification and rationale in the introduction for their focus on cancer screening.

5.       On lines 162-165, there is the indication that the resampled data was seemingly less than the full data.  Some additional explanation of this would be helpful.

6.       The biggest concern with this paper is the definition of “disability” as someone who answered “Yes” to the question “Do you need any support or supervision from others due to your disability or declining physical function?”  This is a very restrictive definition.  There are many people with a documented or diagnosed disability who do not need supports or supervision from others.  There are still others who need that support (or would benefit from it) but refuse to acknowledge it. And there are others who have a documented disabling condition or diagnosis who do not see themselves as having any disability.  Finally, there are some people who have a disability but the type of disability (e.g., frontal lobe brain damage) prevents them from recognizing their disability.  Although the authors somewhat recognize their definition of disability as having implications for their results (in Section 4.4), they need to elaborate upon this limitation.  The major concern here is the likelihood that this definition has already supposed some challenges in managing restrictions of some sort and hence “stacked the deck” to thereby create the likelihood of finding the results they did.  Also, the definition of disability does not take into account those who might have a temporary disabling condition (such as a broken leg).  The authors must address all of these matters in the discussion in some detail.  Readers might mistakenly conclude that this study erroneously indicates that a disability does definitively negatively impact health screening participation.  Although it is possible (albeit, perhaps even likely) this is the case, there needs to be some tempering of the results because of this definition of disability used.

7.       Just a quick note: Table S1 was not available for me to review.    

8.       Line 209; why did the authors group the variable age, or did the information come grouped from the database?  This needs explained. If the authors are the ones who established the age groupings, why?  By doing so, they eliminated the possibility of finding variability within those groupings.  Also, if they are the ones who created these age groupings, on what basis did they create these particular groupings, especially with the unequal number of years for the different groupings (e.g., 20-39 encompasses 40 years but 40-64 encompasses only 15 years).  What was the rationale for those particular age groupings?

9.       Line 216: What is the definition of “constant” for constant visit? 

10.   In Figure 1 (page 7), the Excluded box needs explained.  Were there a particular number or participants excluded because of missing data, etc.?  Or is this the step that got to the particular numbers for each of the blocks detailed after the Data for Analysis block.  Please clarify.

11.   In Tables 1A and 1B, the percentages provided are not the most useful ones.  Instead of identifying the percentages of each sample that were male and female within each grouping (i.e., participants versus non-participants), it would be much more useful (and interesting) to see the percentage of males who were participants and non-participants, the percentage of females who were participants and non-participants, and so on.

12.   The relatively small number of people identified as needing any support or supervision (especially in relation to those who did not identify such a need) raises concerns about comparing data from groups with such disparate numbers of participants.  Although there is not a lot they can do about that, this is something they need to address in the discussion as potentially affecting their results.

13.   There also needs to be increased explanation of the numbers in Table 2A and 2B, especially as related to the confounding variables.  Although experienced statisticians will understand these numbers, others who also might have an interest in this paper might not.

14.   The authors need to make it clear that Figure 2 represents data only from those with a disability as defined in this study.

15.   The discussion is very light on references.  The authors ought to consider relating findings from their study to other studies and implications of those comparisons.  Although they did that a little in section 4.2, this needs expanded, and the other sections really have no such comparisons.

16.   There needs to be some consideration to the title because it is somewhat misleading.  This study is not as much about disability and its impact on screenings as much as it is about individuals who need support because of a disability and that impact on screenings.  This is quite a different question.

In summary, although this study has the potential to contribute to the scientific literature, there need to be several changes to clarify matters and eliminate any misleading attributions. 

Comments on the Quality of English Language

See general comment number 1.  

Author Response

Reply to reviewer 2

Thank you very much for your review and for your helpful comments. We have revised this manuscript on the basis of your comments for us. Revisions are highlighted in yellow color in the manuscript. Please find attached file. 
